# Comparative Genetic Characterization of CTX-M-Producing *Escherichia coli* Isolated from Humans and Pigs with Diarrhea in Korea Using Next-Generation Sequencing

**DOI:** 10.3390/microorganisms11081922

**Published:** 2023-07-28

**Authors:** Kwang-Won Seo, Kyung-Hyo Do, Wan-Kyu Lee

**Affiliations:** 1College of Veterinary Medicine, Chungbuk National University, Cheongju 28644, Republic of Korea; vetskw16@cbnu.ac.kr (K.-W.S.);; 2GutBiomeTech, Cheongju 28644, Republic of Korea

**Keywords:** pathogenic *E. coli*, pig, humans, third-generation cephalosporin, CTX-M

## Abstract

Pathogenic *E. coli* causes intra- and extraintestinal diseases in humans and pigs and third-generation cephalosporins are the primary option for the treatment of these diseases. The objective of this study was to investigate the characteristics and correlation between CTX-M-producing *E. coli* from humans and pigs regarding CTX-M-producing *E. coli* using next-generation sequencing and bioinformatic tools. Among the 24 CTX-M–producing *E. coli*, three types of CTX-M genes (CTX-M-12, CTX-M-14, and CTX-M-15) were detected in humans and four types of CTX-M genes (CTX-M-14, CTX-M-15, CTX-M-55, and CTX-M-101) were detected in pigs. A total of 24 CTX-M–producing *E. coli* isolates also showed the following antimicrobial resistance genes: other B-Lactam resistance gene (75.0%); aminoglycoside resistance genes (75.0%); phenicol resistance genes (70.8%); tetracycline resistance genes (70.8%); sulfonamide resistance genes (66.7%); quinolone resistance genes (62.5%); trimethoprim resistance genes (54.2%); and fosfomycin resistance genes (8.3%). FII (92.3%) and FIB (90.9%) were the most common plasmid replicon in humans and pigs, respectively. A total of thirty-eight different genes associated with virulence 24 CTX-M–producing E. coli and all isolates contained at least more than one virulence gene. A total of 24 CTX-M–producing *E. coli* isolates showed 15 diverse sequence types (STs): thirteen isolates from human belonged to 6 different STs, and 11 isolates from pig belonged to 9 different STs. The presence of virulence genes in *E. coli* together with antimicrobial resistance genes (including CTX-M genes) emphasizes the necessity of comprehensive surveillance and persistent monitoring of the food chain to avoid all types of bacterial contamination, regardless of human or pig origin.

## 1. Introduction

*Escherichia coli* is a widespread and abundant bacterium that forms part of the intestinal commensal microbiota of animals and humans. Although many *E. coli* strains are harmless commensals, the species comprises several zoonotic pathovars that cause intra- and extraintestinal diseases in humans and animals, such as diarrhea, urinary tract infections, meningitis, or septicemia [1]. Fluoroquinolones, trimethoprim–sulfamethoxazole, and third-generation cephalosporins are the primary antimicrobial options for treating infections caused by pathogenic *E. coli* [2].

Third-generation cephalosporin resistance is usually mediated by extended-spectrum β-lactamases (ESBLs), which are generally encoded by genes on mobile genetic elements, such as plasmids and transposons [3]. There are three main types of ESBL (TEM, SHV, or CTX-M) based on their substrate profiles and primary sequences. [4]. Among these, the most prevalent extended-spectrum enzymes correspond to the CTX-M type [5]. Because of the explosive dissemination of CTX-Ms globally, a “CTX-M pandemic” has been reported due to their worldwide increase [6]. CTX-M types have been divided into at least five groups based on amino acid sequence identities CTX-M-1, M-2, M-8, M-9, and M-25. Native CTX-Ms are cefotaximases that usually hydrolyze cefotaxime rather than ceftazidime, but point mutations can extend hydrolysis reactions to ceftazidime and β-lactams [7,8]. In particular, CTX-M β-lactamases are co-associated with resistance to other classes of antimicrobials and high multidrug resistance [8].

Pigs are considered one of the reservoirs of third-generation cephalosporin-resistant pathogenic *E. coli* found in food, such as pork products, and they can transfer their resistance and virulence genes to human pathogens via the human gut through the food chain [9]. Although many antimicrobial resistance studies have been conducted in humans and pigs, little is known about the interrelationships between humans, pigs, and CTX-M-producing pathogenic *E. coli*. Therefore, this study aimed to investigate the characteristics of CTX-M-producing *E. coli* and compare them between humans and pigs using next-generation sequencing and bioinformatics tools.

## 2. Materials and Methods

### 2.1. Bacterial Strains

Among the 589 pathogenic *E. coli* isolates (392 isolates from 401 pigs with colibacillosis on 120 different farms, and 197 isolates from patients with diarrhea provided by the National Culture Collection for Pathogens, Gyeongsang National University Hospital Branch of the NCCP and Kyungpook National University Hospital Branch of the NCCP) previously described [10], 103 third-generation cephalosporin-resistant *E. coli* isolates were analyzed to confirm the presence of the CTX-M gene by applying a previously published PCR protocol [9,11]. Subsequently, 24 third-generation cephalosporin-resistant *E. coli* isolates (from humans: 13 isolates; from pigs: 11 isolates) were shown to carry CTX-M and were analyzed in this study.

### 2.2. DNA Extraction and Quantification

Genomic DNA from 24 CTX-M-producing *E. coli* was extracted using the Maxwell RSC Instrument (Promega, Madison, WI, USA) and Maxwell^®^ RSC Cultured Cells DNA Kit AS1620 (Promega), according to the manufacturer’s instructions. DNA was quantified using Nano-Drop Spectrophotometer (Thermo Scientific, Waltham, MA, USA) and a Qubit 4 fluorometer (Thermo Scientific, Waltham, MA, USA) according to the manufacturer’s protocol. DNA was stored at −20 °C until sequencing.

### 2.3. Whole Genome Sequencing and Analysis

Paired-end DNA libraries were prepared using a Nextera kit (Illumina, San Diego, CA, USA) according to the manufacturer’s instructions. Briefly, genomic DNA was tagmented by simultaneously fragmenting and tagging with adapter sequences using the Nextera transposome (Nextera XT DNA Library Preparation Kit, Illumina, San Diego, CA, USA). The tagmented DNA was then amplified using a PCR program. The amplified DNA was purified using with AMPure XP beads. Nextera libraries were then quantified using a Qubit 4 fluorometer, and the size profile was evaluated on an Agilent Technology 2100 Bioanalyzer (Agilent Technologies, Waldbronn, Germany). Sequencing was performed on the Illumina MiSeq instrument in a 2 × 300 bp format using a MiSeq reagent kit v3 (Illumina). The SPAdes (v3.12) assembler was used for generating assembled genomes and analyzed for antibiotic resistance characteristics (ResFinder v3.1), sequence type (ST) (MLST v2.0), virulence genes (VirulenceFinder v2.0), and incompatibility groups (Inc types) of plasmids (PlasmidFinder v2.0) from the Center for Genomic Epidemiology (Appendix A).

## 3. Results

### 3.1. Characteristics of CTX-M-Producing E. coli

Regarding the CTX-M type distribution, the most frequent CTX-M genes in humans were CTX-M-15 (n = 11), followed by CTX-M-12 (n = 1) and CTX-M-14 (n = 1), while those in pigs were CTX-M-14 (n = 4) and CTX-M-55 (n = 4), followed by CTX-M-15 (n = 2) and CTX-M-101 (n = 1). Among the 24 CTX-M-producing *E. coli* isolates, β-lactam, TEM-1, OXA-1, NDM-5, and CMY-2 genes were detected in 15, 9, 2, and 2 isolates, respectively. In addition to the β-lactam genes, the isolates also contained genes conferring resistance to aminoglycoside, *aac(3)-Ila*, *aac(3)-Ild*, *aac(3)-IV*, *aadA1*, *aadA2*, *aadA5*, *aadA22*, *aadA25*, *aph(3*″*)-Ia*, *aph(3*″*)-Ib*, *aph(4)-Ia*, and *aph(6)-Id* (humans: 10 isolates, pigs: 8 isolates); phenicol, *catA2*, *catB3*, *cmlA1*, and *floR* (humans: 8 isolates, pigs: 9 isolates); quinolone, *aac(6*′*)-Ib-cr*, *qacE*, and *qnrS1* (humans: 11 isolates, pigs: 4 isolates); sulfonamide, *sul1*, *sul2*, and *sul3* (humans: 7 isolates, pigs: 9 isolates), tetracycline, *tet(A)*, and *tet(B)* (humans: 8 isolates, pigs: 9 isolates); trimethoprim, *dfrA12*, *dfrA14*, *dfrA15*, and *dfrA17* (humans: 7 isolates, pigs: 6 isolates); and fosfomycin, *fosA3* (humans: 1 isolate, pigs: 1 isolate) in both humans and pigs. The genes *mph(A)* and *erm(B)* were only detected in the human isolates. In particular, amino acid substitutions in the quinolone resistance-determining regions (QRDR) of GyrA (S83L, D87N), ParC (S80I, S80R, E84V), and ParE (S458A, I529L) were found in 12 isolates (92.3%) from humans and 7 isolates (63.6%) from pigs. Twelve different plasmid replicon types were found in different isolates and in different proportions in humans and pigs. In humans, FII (n = 12) was the most predominant plasmid replicon type detected, and over 90% of isolates harbored two or more plasmids (Figure 1). In pigs, FIB (n = 10) was the most predominant plasmid replicon type detected, and all the isolates harbored two or more plasmids (Figure 1).

### 3.2. Virulence Factors

A total of 38 different genes associated with virulence were identified in 24 CTX-M-producing *E. coli* isolates, and all isolates contained at least more than one virulence gene. The genes *iha* and *papA* (coding for adhesins), *fyuA*, and *irp2* (coding for iron acquisition systems), *kpsMII* (coding for adhesins), and *sat* (coding for toxins) were the most frequent virulence genes in CTX-M-producing *E. coli* isolates from humans. The genes *csgA* (coding for adhesins), *iucC*, *iutA*, and *sitA* (coding for iron acquisition systems); *traT* (coding for adhesins); and *hlyE* (coding for toxins) were the most frequent virulence genes in CTX-M-producing *E. coli* isolates from pigs. The *ompT* gene was the most predominant virulence gene coding for miscellaneous factors in both humans and pigs.

### 3.3. Multi-Locus Sequence Typing (MLST)

The 24 sequenced CTX-M-producing *E. coli* isolates belonged to 15 diverse STs (Figure 2). Thirteen isolates from humans belonged to six different STs, namely, ST131 (n = 6), ST410 (n = 3), ST38 (n = 1), ST88 (n = 1), ST405 (n = 1), and ST648 (n = 1). Eleven isolates from pigs belonged to nine different STs, namely, ST10 (n = 2), ST100 (n = 2), ST101 (n = 1), ST224 (n = 1), ST327 (n = 1), ST457 (n = 1), ST542 (n = 1), ST1642 (n = 1), and ST2216 (n = 1). The same STs did not exist in both humans and pigs.

## 4. Discussion

In this study, three types of CTX-M genes (CTX-M-12, CTX-M-14, and CTX-M-15) were detected in humans and four types of CTX-M genes (CTX-M-14, CTX-M-15, CTX-M-55, and CTX-M-101) were detected in pigs. CTX-M-14 and CTX-M-15 were identified in both humans and pigs. They have been reported as the dominant CTX-M type in livestock, including healthy animals and retail meats in China [12], the United Kingdom [13], Spain [14], and Japan [15]. CTX-M-15, which is frequently associated with humans worldwide, has recently been reported as the dominant ESBL in clinical *E. coli* in humans, and it was the most frequent CTX-M gene (54.2%) in this study. CTX-M-producing *E. coli* carrying the CTX-M-12 gene has also been reported as a common CTX-M type among Enterobacteriaceae [16,17]. In addition, CTX-M-55 and CTX-M-101, with only a 2-substitution difference from CTX-M-14, have often been isolated from livestock and humans in other countries [18,19,20,21,22]. In this study, TEM-1 and OXA-1, which encode other enzymes that confer β-lactam resistance, were identified in 15 CTX-M-producing *E. coli* isolates (human: 8 isolates, pigs: 7 isolates) and 9 CTX-M-producing *E. coli* isolates (human: 8 isolates, pigs: 1 isolate), respectively. These genes are not ESBLs; however, they can be induced into ESBLs by mutations that alter the amino acid sequence around the active site [23].

Third-generation cephalosporin-resistant *E. coli* may have been selected and maintained in humans and pigs owing to the use of third-generation cephalosporins or co-selection after the use of other antimicrobials, such as aminoglycosides, β-lactams, phenicol, quinolone, sulfonamides, trimethoprim, and tetracyclines [3]. The majority of isolates described in this study also contained resistance genes to these classes of antimicrobials. The *tetA* gene was one of the most frequently observed resistant genes in the CTX-M-producing *E. coli* isolates, which is consistent with the findings of Girlich et al. [24]. The sul1 and sul2 genes, which encode a sulfonamide-resistant dihydropteroate synthase, were identified in 11 (45.8%) CTX-M-producing *E. coli* isolates. In addition, we found that 12 (50.0%) CTX-M-producing *E. coli* isolates carried *aph(3*″*)-Ib* and *aph(6)-Id* genes, which encode aminoglycoside adenylyltransferases. These two genes have already been reported as major determinants of sulfonamide and gentamicin resistance in pathogenic *E. coli*, respectively [25,26]. Although chloramphenicol was banned in food-producing animals in Korea in 1990 because of its suspected carcinogenicity, *catB3* and *cmlA1* genes, which encode a specific chloramphenicol transporter, were detected in nine (37.5.0%) and five (20.8%) CTX-M-producing *E. coli* isolates, respectively. The plasmid-mediated quinolone resistance (PMQR) genes *aac(6*′*)-Ib-cr*, *qacE*, and *qnrS1* were identified in both pigs and humans. PMQR genes may be significantly associated with the β-lactamase gene and are detected in CTX-M-producing *E. coli* at high levels [27]. In addition, the most common plasmid replicons were IncF plasmids, including FIB (79.2%), FII (70.8%), and FIA (62.5%). IncF plasmids play an important role in the spread of antimicrobial (including ESBLs) and virulence and resistant determinants among pathogenic *E. coli* [28].

Some virulence-associated genes have been described as important genes because they cause intestinal/extraintestinal infections in humans and animals [29]. The *air*, *astA*, and *eilA* genes, which were detected in pig isolates, have been related with enteroaggregative *E. coli* in humans [30]. These virulence genes can play a role in the pathogenic potential in humans [30]. The *iss* and *lpfA*, associated with the extraintestinal pathogenic *E. coli* reported in a humans and pigs worldwide, were also found in both humans and pigs in this study [31,32,33]. In particular, the *iss* gene is recognized as being liable for *E. coli* immune evasion by increasing serum survival and also for its role in extraintestinal pathogenic *Escherichia coli* (ExPEC) for enhanced survival of bacteria in serum [30]. In addition, the *iha* gene, dominating the ExPEC adherence virulence genes, has been identified in both humans and pigs [34,35]. We also found several toxins that belong to the ExPEC group, such as *ireA*, *cnf1*, *vat*, *sat*, *senB*, and *pic*, in animals, humans, or both.

In this study, CTX-M-15-producing E. coli was mainly related with the ST131, which was not found in the E. coli from pigs in our study. CTX-M-15 was detected in *E. coli* ST327 and ST1642 in pigs, which were also found in human patients [36,37] but less frequently than ST131. The most prevalent STs in pigs were ST100 and ST10, which are the predominant enterotoxigenic *E. coli* types, and which are important pig pathogens in Canada, Germany, the United States, and Thailand (http://mlst.warwick.ac.uk/mlst/dbs/Ecoli, accessed on 30 April 2023). Because diverse STs are related to pathogenic and antimicrobial resistant strains, the emergence of a pathogenic *E. coli* showing diverse STs may pose a risk.

## 5. Conclusions

In this study, we genetically characterized and investigated the correlation of CTX-M-producing *E. coli* isolated from humans and pigs suffering from diarrhea in Korea. Although diseased pigs are less likely to be a source of antimicrobial-resistant bacteria compared to healthy pigs that enter food chain, the emergence of antimicrobial-resistant bacteria can be a public health problem as they are transmitted between pigs. This is the first study to genetically characterize and investigate the prevalence and interrelation of CTX-M-producing *E. coli* isolated from humans and pigs with diarrhea in Korea. The presence of virulence genes in *E. coli* together with antimicrobial resistance genes (including CTX-M genes) emphasizes the necessity for comprehensive surveillance and persistent monitoring of the food chain to avoid all types of bacterial contamination, irrespective of human or pig origin.

## Figures and Tables

**Figure 1 microorganisms-11-01922-f001:**
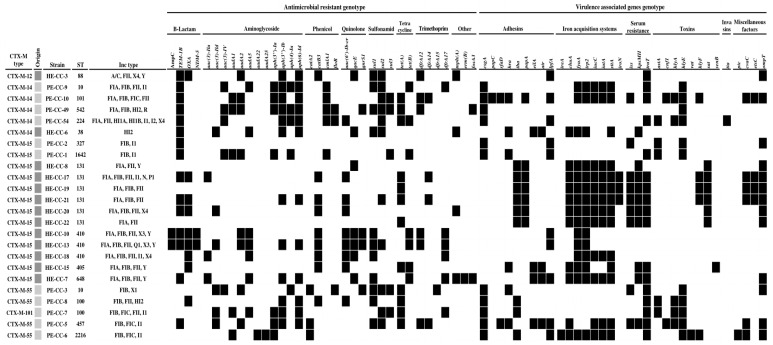
Molecular characteristics of the 24 CTX-M-producing *Escherichia coli* isolated from humans and pigs in Korea. Dark gray squares indicate isolates from humans; light gray squares indicate isolates from pigs; black squares indicate positivity for genes.

**Figure 2 microorganisms-11-01922-f002:**
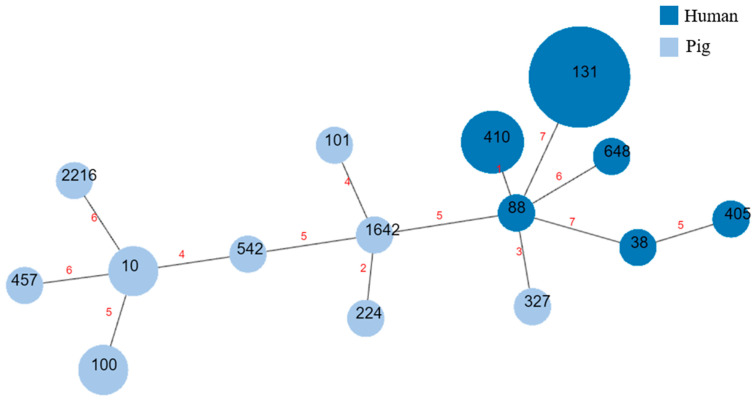
Minimum spanning tree based on the sequence type (ST) of 24 CTX-M-producing *Escherichia coli* isolated from humans and pigs in Korea. Every circle represents an ST (the ST number is shown in the circle), and the size of the circle represents the number of isolates.

## Data Availability

Data are contained within the article.

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
