# Peer review of "Comparative Genetic Characterization of CTX-M-Producing Escherichia coli Isolated from Humans and Pigs with Diarrhea in Korea Using Next-Generation Sequencing"

_microorganisms, 2023, doi:10.3390/microorganisms11081922_

Round 1

Reviewer 1 Report

 This study aimed to investigate and characterize CTX-M-producing E. coli isolated from humans and pigs with diarrhoea in Korea and comparing between the isolates using next-generation and bioinformatics tools.  However, the study is properly designed and the materials address the purpose of the study, there are some minor comments that should be modified.
-    I suggest that the manuscript is suitable for publication after minor revision.
  Minor comments
-    Please revise the manuscript for English language and carefully correct sentences related mistakes and grammatical errors.
-    Line 9: replace cause with causes
-    Line 15: replace resistant with resistance
-    Line 21: replace isolate with isolates
-    Line 25: insert (of ) between the necessity comprehensive 
-    Line 203: The sentence should be written as : In this study, we genetically characterized and investigated the prevalence and ………………….etc.

Please correct some grammatical and typing errors in the attached PDF  

n

Author Response

Thank you for your review. I proceeded with the revision carefully according to your suggestion.

Following changes are suggested:

*Comment: Line 9: replace cause with causes.

Response: As your suggestion, I revised the word.

*Comment: Line 15: replace resistant with resistance.

Response: As your suggestion, I revised the word.

*Comment: Line 21: replace isolate with isolates.

Response: As your suggestion, I revised the word.

*Comment: Line 25: insert (of) between the necessity comprehensive.

Response: As your suggestion, I added the word (of).

*Comment: Line 203: The sentence should be written as : In this study, we genetically characterized and investigated the prevalence and ………………….etc.

Response: As your suggestion, I revised this sentence.

Reviewer 2 Report

Authors have compared CTX-M E.coli from humans and pigs in Korea. The topic is well-suited for this journal and has a public health implication. Here are my suggestions:

1. Line 50: Pigs are one of the reservoirs and not the primary reservoirs of ESBLS.

2. Was there QA performed on genome assemblies?What about trimming?

3. Authors have compared isolates from diseased pigs.Diseased pigs are less likely to be a source of AMR bacteria as compared to healthy pigs that enter food chain. Please consider adding such isolates or note this as a limitation in discussion.

4. Although not perfect, but MOB-SUITE can be used to assemble plasmids. This can then tell you if plasmids carrying CTX genees were similar in humans and pigs. It is often noted that the bacterial ST between humans and animals do not match, but plasmids do!

5. Figure 1 is hard to read. Consider splitting in 2.

6. Please consider uploading raw reads to NCBI.

2.

Author Response

Thank you for your review. I proceeded with the revision carefully according to your suggestion.

*Comment: 1. Line 50: Pigs are one of the reservoirs and not the primary reservoirs of ESBLS.

Response: As your suggestion, we revised the sentence as follows.

- Page 2, Line 50: ‘Pigs are considered one of the reservoirs of third-generation cephalosporin-resistant pathogenic E. coli

*Comment: 2. Was there QA performed on genome assemblies? What about trimming?

Response: SPAdes was used for genome assemblies and low quality genomes were trimmed.

*Comment: 3. Authors have compared isolates from diseased pigs. Diseased pigs are less likely to be a source of AMR bacteria as compared to healthy pigs that enter food chain. Please consider adding such isolates or note this as a limitation in discussion.

Response: As your suggestion, we added the sentence as follows.

- Page 5-6, Line 205-208: ‘Although diseased pigs are less likely to be a source of antimicrobial-resistant bacteria as compared to healthy pigs that enter food chain, the emergence of antimicrobial-resistant bacteria can be a public health problem as they are transmitted between pigs.’

*Comment: Although not perfect, but MOB-SUITE can be used to assemble plasmids. This can then tell you if plasmids carrying CTX genees were similar in humans and pigs. It is often noted that the bacterial ST between humans and animals do not match, but plasmids do!

Response: I totally agree your opinion (It is often noted that the bacterial ST between humans and animals do not match, but plasmids do). As your suggestion, I used MOB-SUITE for plasmid assembly, but it's not works well. Through additional experiments, we have plane have a plan to DNA sequencing in a long-read method. And then the whole-genome-sequencing will be carried out in a hybrid method, and the plasmid will be assembled thereafter.

*Comment: Figure 1 is hard to read. Consider splitting in 2.

Response: Thank you for your suggestion. In my opinion, to check the molecular characteristics of the 24 CTX-M-producing Escherichia coli at once, it seems better not to split the Figure.

*Comment: Please consider uploading raw reads to NCBI.

Response: We have already uploaded raw data to NCBI, and it can be found in 'Supplementary material (Data availability in NCBI)'.

Reviewer 3 Report

This is a communication about the characteristics of CTX-M-producing E. coli isolated from humans and pigs with diarrhea and the correlation between them. According to authors this is the first study to genetically characterize and investigate the interrelation of CTX-M-producing E. coli isolated from humans and pigs.

In Introduction, saying that “ESBLs can be grouped into three main types” may induce a less knowledgeable reader to consider that these are the only groups of ESBLs that exist. It is recommended to reformulate the sentence.

Still in the introduction and due to the fact that there are already authors who consider that there are more than 5 groups of CTX-M, We suggest in Line 45 - “…divided into at least five groups based on amino acid sequence identities:…” instead of “…divided into five groups based on amino acid sequence identities:…”

In Material and Methods authors refer that “Among the 589 pathogenic E. coli isolates…previously described, 103 third-generation cephalosporin-resistant E. coli isolates were analyzed to confirm the presence of the CTX-M gene”. Considering that authors worked with bacterial isolates, Why have they used the Maxwell® RSC Fecal Microbiome DNA Kit with the Maxwell® RSC Instruments, a method for isolation of nucleic acid from fecal samples?

Line 53 – Reference [10] appears before reference [9] in line 63

Line 107 – All antibiotics are in lowercase, so fosfomycin should also be in lowercase.

Line 150 – Please check if the value 84.6% is correct. I think it is 54.2% if you want to refer to all the 24 CTX-M-producing E. coli studied.

Line 164 – According to Table 1 gene tetA was detected in 12 CTX-M-producing E. coli isolates but there are other genes that were also detected in the same number of isolates, so it was not the most frequently observed resistant gene, but one of the most frequently observed….

Line 164 – tetA in italics

Line 189 - Acronyms and abbreviations should be defined the first time they appear in the text: Extraintestinal pathogenic Escherichia coli (ExPEC)

In this study the focus was not the study of prevalence so, we suggest authors that in conclusions should say “…we analyzed, genetically characterized, and investigated the correlation of CTX-M-producing E. coli isolated from humans and pigs suffering from diarrhea in Korea….” Instead of “…we analyzed genetically characterize and investigate the prevalence and correlation of CTX-M-producing E. coli isolated from humans and pigs suffering from diarrhea in Korea.”

A review of English by a native speaker is recommended regardless of the suggestions mentioned below:

Line 9-10 – We suggest: “Pathogenic E. coli cause intra- and extraintestinal diseases in humans and pigs and third-generation cephalosporins are primary option for the treatment of these diseases.” Instead of “Pathogenic E. coli cause intra- and extraintestinal diseases in humans and pigs and third-generation cephalosporin is primary option for the treatment of this disease.”

Line 11-12 - We suggest:“…correlation between CTX-M-producing E. coli from humans and pigs…” instead of “…correlation between humans and pigs about CTX-M-producing E. coli…”

Line 15 – We suggest: “Also, 24 CTX-M–producing E. coli isolates showed the following antimicrobial resistance genes:…” instead of “Also, 24 CTX-M–producing E. coli isolates showed antimicrobial resistant characteristics as followed:…”

Line 21 – “…were identified in 24 CTX-M–producing E. coli and all isolates contained at least more than one virulence gene.” Instead of “…were identified in 24 CTX-M–producing E. coli and all isolate contained at least one more virulence gene.”

Line 22 – “…showed 15 diverse sequence types (STs)…” instead of “…showed to 15 high diversity of sequence types (STs)…”

Line 25 – “…the necessity of a comprehensive surveillance…” instead of “…the necessity comprehensive surveillance..”

Line 108 – “The genes mph(A) and erm(B) were only detected in the human isolates.” Instead of “Only mph(A) and erm(B) were detected in the human isolates.

Line 123 – “…all isolates contained at least more than one virulence gene…” instead of “…all isolates contained at least one more virulence gene…”

Line 153 – “…isolated from livestock…” instead of “…iso-lated from livestock…”

Author Response

Thank you for your review. I proceeded with the revision carefully according to your suggestion.

*Comment: In Introduction, saying that “ESBLs can be grouped into three main types” may induce a less knowledgeable reader to consider that these are the only groups of ESBLs that exist. It is recommended to reformulate the sentence.

Response: As your suggestion, we revised the sentence as follows.

- Page 1, Line 40-41: ‘There are three main types of ESBL (TEM, SHV, or CTX-M) based on their substrate profiles and primary sequences’

*Comment: Still in the introduction and due to the fact that there are already authors who consider that there are more than 5 groups of CTX-M, We suggest in Line 45 - “…divided into at least five groups based on amino acid sequence identities:…” instead of “…divided into five groups based on amino acid sequence identities:…”.

Response: As your suggestion, I revised the sentence.

*Comment: In Material and Methods authors refer that “Among the 589 pathogenic E. coli isolates…previously described, 103 third-generation cephalosporin-resistant E. coli isolates were analyzed to confirm the presence of the CTX-M gene”. Considering that authors worked with bacterial isolates, Why have they used the Maxwell® RSC Fecal Microbiome DNA Kit with the Maxwell® RSC Instruments, a method for isolation of nucleic acid from fecal samples?.

Response: We wrote the wrong name of the kit. The kit used here was Maxwell® RSC Cultured Cells DNA Kit AS1620, and this kit was used to DNA extraction. we revised the name of the kit in Line 71-72.

*Comment: Line 53 – Reference [10] appears before reference [9] in line 63.

Response: As your suggestion, I revised Reference 9 and 10.

*Comment: Line 107 – All antibiotics are in lowercase, so fosfomycin should also be in lowercase.

Response: As your suggestion, I revised the word.

*Comment: Line 150 – Please check if the value 84.6% is correct. I think it is 54.2% if you want to refer to all the 24 CTX-M-producing E. coli studied.

Response: Your opinion is correct. I revised the number.

*Comment: Line 164 – According to Table 1 gene tetA was detected in 12 CTX-M-producing E. coli isolates but there are other genes that were also detected in the same number of isolates, so it was not the most frequently observed resistant gene, but one of the most frequently observed….

Response: I agree your opinion. As your suggestion, I revised the sentence.

*Comment: Line 164 – tetA in italics

Response: As your suggestion, I revised the word.

*Comment: Line 189 - Acronyms and abbreviations should be defined the first time they appear in the text: Extraintestinal pathogenic Escherichia coli (ExPEC)

Response: As your suggestion, I added ‘Extraintestinal pathogenic Escherichia coli (ExPEC)’ in Line 189.

*Comment: In this study the focus was not the study of prevalence so, we suggest authors that in conclusions should say “…we analyzed, genetically characterized, and investigated the correlation of CTX-M-producing E. coli isolated from humans and pigs suffering from diarrhea in Korea….” Instead of “…we analyzed genetically characterize and investigate the prevalence and correlation of CTX-M-producing E. coli isolated from humans and pigs suffering from diarrhea in Korea.”

Response: As your suggestion, I revised the sentence as follows.

- Page 5-6, Line 204-205: ‘we genetically characterized and investigated the correlation of CTX-M-producing E. coli isolated from humans and pigs suffering from diarrhea in Korea.’

*Comment: Line 9-10 – We suggest: “Pathogenic E. coli cause intra- and extraintestinal diseases in humans and pigs and third-generation cephalosporins are primary option for the treatment of these diseases.” Instead of “Pathogenic E. coli cause intra- and extraintestinal diseases in humans and pigs and third-generation cephalosporin is primary option for the treatment of this disease.”

Response: As your suggestion, I revised the sentence.

*Comment: Line 11-12 - We suggest:“…correlation between CTX-M-producing E. coli from humans and pigs…” instead of “…correlation between humans and pigs about CTX-M-producing E. coli…”

Response: As your suggestion, I revised the sentence.

*Comment: Line 15 – We suggest: “Also, 24 CTX-M–producing E. coli isolates showed the following antimicrobial resistance genes:…” instead of “Also, 24 CTX-M–producing E. coli isolates showed antimicrobial resistant characteristics as followed:…”

Response: As your suggestion, I revised the sentence.

*Comment: Line 21 – “…were identified in 24 CTX-M–producing E. coli and all isolates contained at least more than one virulence gene.” Instead of “…were identified in 24 CTX-M–producing E. coli and all isolate contained at least one more virulence gene.”

Response: As your suggestion, I revised the sentence.

*Comment: Line 22 – “…showed 15 diverse sequence types (STs)…” instead of “…showed to 15 high diversity of sequence types (STs)…”

Response: As your suggestion, I revised the sentence.

*Comment: Line 25 – “…the necessity of a comprehensive surveillance…” instead of “…the necessity comprehensive surveillance.”

Response: As your suggestion, I revised the sentence.

*Comment: Line 108 – “The genes mph(A) and erm(B) were only detected in the human isolates.” Instead of “Only mph(A) and erm(B) were detected in the human isolates.

Response: As your suggestion, I revised the sentence.

*Comment: Line 123 – “…all isolates contained at least more than one virulence gene…” instead of “…all isolates contained at least one more virulence gene…”

Response: As your suggestion, I revised the sentence.

*Comment: Line 153 – “…isolated from livestock…” instead of “…iso-lated from livestock…”

Response: As your suggestion, I revised the sentence.